# Hypoxic Neuroinflammation in the Pathogenesis of Multiple Sclerosis

**DOI:** 10.3390/brainsci15030248

**Published:** 2025-02-26

**Authors:** Bethany Y. A. Hollingworth, Patrick N. Pallier, Stuart I. Jenkins, Ruoli Chen

**Affiliations:** 1School of Allied Health Professions and Pharmacy, Keele University, Staffordshire ST5 5BG, UK; b.hollingworth@keele.ac.uk; 2Centre for Neuroscience, Surgery and Trauma, The Blizard Institute, Barts and The London School of Medicine and Dentistry, Queen Mary University of London, London E1 2AT, UK; p.pallier@qmul.ac.uk; 3Neural Tissue Engineering Keele (NTEK), School of Medicine, Keele University, Staffordshire ST5 5BG, UK; s.i.jenkins@keele.ac.uk

**Keywords:** multiple sclerosis, demyelination, remyelination, neuroinflammation, immunomodulation, neurodegeneration, hypoxia, HIF, HB-EGF

## Abstract

Multiple sclerosis (MS) is an autoimmune disease that damages the myelin sheath around the central nervous system axons, leading to neurological dysfunction. Although the initial damage is driven by inflammation, hypoxia has been reported in several brain regions of MS patients, but the significance of this for prognosis and treatment remains unclear. Neuroinflammation can induce hypoxia, and hypoxia can induce and exacerbate neuroinflammation, forming a vicious cycle. Within MS lesions, demyelination is often followed by remyelination, which may restore neurological function. However, demyelinated axons are vulnerable to damage, which leads to the accumulation of the permanent neurological dysfunction typical in MS, with this vulnerability heightened during hypoxia. Clinically approved therapies for MS are immunomodulatory, which can reduce relapse frequency/severity, but there is a lack of pro-regenerative therapies for MS, for example promoting remyelination. All tissues have protective responses to hypoxia, which may be relevant to MS lesions, especially during remyelinating episodes. When oxygen levels are reduced in the brain, constitutively expressed hypoxia-inducible factors (HIF) are stabilised, upregulating hundreds of genes, including neuroprotective factors. Furthermore, astrocytes upregulate heparin-binding epidermal growth factor (EGF)-like growth factor (HB-EGF) in the early stage of MS. HB-EGF promotes protective mechanisms and induces oligodendrocyte and neuron differentiation and survival. This review article outlines the neuroinflammation and hypoxia cycle in MS pathology and identifies potential therapeutic targets to limit neurodegeneration and/or promote regeneration. Both HIF and HB-EGF signalling pathways induce endogenous protection mechanisms in the CNS, promoting neuroprotection and remyelination directly, but also indirectly by modulating the immune response in MS. Promoting such endogenous protective signalling pathways could be an effective therapy for MS patients.

## 1. Introduction

Multiple sclerosis (MS) is a chronic illness that affects the central nervous system (CNS). It occurs when the immune system mistakenly attacks the protective coating of nerve fibres called myelin, leading to inflammation and damage [1]. This can cause a wide range of symptoms, including fatigue, difficulty with walking, numbness or tingling in limbs, vision problems, and problems with coordination and balance [2]. The exact cause of MS is still not completely understood, but a combination of genetic and environmental factors is believed to contribute to its development. While MS affects millions globally [1], its prevalence is higher in regions farther from the equator, and it is more common in women than men [3,4]. The highest prevalence of MS in the world is in northern Europe, where the suspected genes of MS were brought from the east around 5000 years ago [3]. One of the suspect genes is HLA-DRB1*1501, which encodes antigen-presenting proteins [5].

In MS, the myelinated axons in the CNS are attacked by the body’s own immune system, leading to progressive demyelination of axons that build up in the degeneration of denuded axons. One of the key features of MS is demyelinated plaques in both grey and white matters in the CNS [6,7]. Histopathological results of early demyelinating lesions in MS show an inflammatory infiltrate surrounding small veins with monocytes, macrophages, and invading lymphocytes [6,7]. Hypoxia is evidenced by hypoxia inducible factor 1α (HIF-1α) expression in experimental autoimmune encephalomyelitis (EAE) mouse models of MS, and enhanced hypoxia is associated with more significant neurological deficits [8]. There is a strong correlation between the level of hypoxia and the degree of white matter inflammation in EAE mice [9]. MS patients have marked hypoxic regions in the cerebral cortex [10] and hypoxia could persist for at least a year in 80% of MS patients [11].

Current MS therapies, approved by regulatory agencies such as the U.S. Food and Drug Administration (FDA), are effective in reducing relapse rates and managing inflammation during the early relapsing–remitting stages of the disease. However, these treatments have limited efficacy in preventing or halting MS progression, highlighting the urgent need for novel therapeutic strategies aimed at the earlier stages of MS. A better understanding of the biomarkers and mechanisms underlying the early phases of MS could significantly improve early diagnosis, prevention, and treatment outcomes.

This narrative review aims to investigate the roles of the interaction between neuroinflammation and hypoxia in MS progression and identify emerging biomarkers for MS progression.

## 2. Progressive Nature of Multiple Sclerosis

MS is a neurodegenerative and progressive disease in which the condition worsens and for which there is currently no cure [2]. The progressive nature of MS is marked by an initial period of relapsing–remitting symptoms, followed by a gradual worsening of neurological function, with increasing disability over time [12]. The earliest clinical manifestation of MS is often a clinically isolated syndrome (CIS), which represents the first neurological episode triggered by inflammation or demyelination [13]. Although patients with CIS may recover fully or partially from the initial episode, CIS is frequently the first sign of MS, with a substantial proportion of patients eventually progressing to relapsing–remitting MS (RRMS) [14]. RRMS is the most common form of MS, affecting approximately 85–90% of patients. It is characterized by episodic neurological deterioration, or relapses, followed by periods of symptom recovery (remission). While patients experience complete or partial recovery during remission, the underlying disease process continues, and disability gradually accumulates over time [15]. RRMS may eventually transition into secondary progressive MS (SPMS), which is marked by more pronounced disability accumulation and fewer relapses [16]. Approximately 80% of RRMS patients transition to SPMS within 10 years, with the majority progressing to SPMS within 25 years [15].

As the disease progresses, brain atrophy becomes more pronounced, and axon loss is seen leading to irreversible neurological deficit [7]. Irreversible axonal damage is the essential cause of disease progression and non-remitting clinical disability [17].

The number of plaques increases with the disease duration and the age of the patient [18]. In both primary and secondary progressive MS (PPMS and SPMS), there is widespread demyelination in the cerebellar and cerebral cortex and diffuse degenerative changes in grey and white matter [18]. The progressive stage eventually leads to severe atrophy in the CNS with dilation of cerebral ventricles and tissue loss.

## 3. Neuroinflammation and Hypoxia Cycle in Multiple Sclerosis

### 3.1. Neuroinflammation

In MS, the blood–brain barrier (BBB) becomes more permeable, allowing immune cells like T-cells and B-cells, as well as monocytes and macrophages, to invade the CNS, triggering an inflammatory cascade [19]. In both early and progressive MS, inflammation remains central to the development and expansion of lesions, but inflammation is more pronounced in RRMS than in the progressive stages of MS, so anti-inflammation therapies work more effectively in RRMS patients compared to SPMS or PPMS patients [6]. In RRMS, inflammation and demyelination occur in isolated episodes or relapses, followed by partial or complete recovery during remission periods [20]. By contrast, the inflammatory activity is more persistent in progressive MS (both primary and secondary), leading to cumulative neurodegeneration and progressive disability over time [21]. This suggests that while inflammation continues in progressive stages, the processes of demyelination and axonal loss become more dominant as the disease advances. Indeed, recent research highlights that microglial activation continues even in progressive stages, and even in the absence of overt inflammatory cell infiltrates, indicating that neuroinflammation persists and drives damage in both the early and advanced stages of the disease [21,22].

Active demyelination is associated with a rim of activated microglia at the border between normal-appearing grey matter and cortical plaques, while inactive plaques are separated areas of microglia activation and demyelination without activity at the edges [22]. Demyelination affects oligodendrocytes and myelin sheaths, whilst neurons and axons are preserved in the initial stages of lesion formation [23]. Initial tissue injury occurs within a small rim that extends from the margin of the plaque into the surrounding peri-plaque white matter. This is known as the pre-phagocytic stage of MS lesion formation, where mechanisms of neurodegeneration or demyelination can be identified [24]. Smouldering lesions, also known as slowly expanding lesions, which are characterised by chronic inflammation and gradual expansion over time, are a significant factor in the progression of MS [25]. Patients with more active smouldering lesions tend to experience a quicker decline in their physical and cognitive functions [26]. Cortical lesions are seen in all stages of MS but they are sparse in RRMS and more common in the progressive stage. Subpial cortical demyelination is only seen in MS but not in other inflammatory diseases of the CNS [27].

### 3.2. Hypoxia

In MS patients, histological studies have shown evidence of hypoxia within the lesions, such as pattern III demyelination, which is a form of demyelination that occurs under hypoxic conditions [28]. It is characterized by the preferential loss of myelin-associated glycoprotein (MAG), a key component of the myelin sheath that helps in its formation and maintenance [29,30]. This type of demyelination is thought to be closely linked to ischemic damage in the CNS, where a poor oxygen supply leads to local myelin degradation [31]. The use of normobaric oxygen therapy (95% O_2_) in animal models, such as Dark Agouti rats with EAE, has shown promising results, the therapy significantly reducing demyelination in these animals [28]. The demyelination in active EAE is believed to be mediated by anti-myelin oligodendrocyte glycoprotein (MOG) antibodies [32,33], while these lesions are comparable to those seen in Patterns I and II demyelination in MS [34].

In hypoxia, it is the oxidative damage rather than the failure of aerobic respiration that is the lethal underlying mechanism. Oxidative stress increases in hypoxia due to the change of oxidation of nicotinamide adenine dinucleotide—related substrate (complex 1) to succinate-dependent oxidation (complex 2) that occurs in this state [35]. The white–grey matter border and dorsal white column of the spinal cord are key regions where early hypoxia is observed in MS and high amounts of nitric oxide and reactive oxygen species (ROS) are seen [30,31]. The white-grey matter border is particularly significant because it represents a transition zone where white matter meets grey matter. These areas may be more vulnerable to ischemic damage, as they are involved in complex neurovascular interactions [2].

Van den Bosch et al. [20] found that where myelin wrapping was less compact in the optic nerves of MS patients there were more axonal mitochondria, indicating that more energy was needed for signal propagation and maintenance of the nerve fibres. However, more mitochondria inevitably produce ROS, which would be an amplifying factor for myelin breakdown.

Oligodendrocytes are highly vulnerable to oxidative stress because of two main factors: their inability to effectively neutralize free radicals and the presence of large iron stores in these cells, which can promote radical formation [36,37]. Under sustained hypoxic conditions, particularly in an inflammatory environment, oligodendrocytes undergo selective death, leading to a form of demyelination known as Pattern III, which is akin to the processes seen in MS [31,34,38,39]. Interestingly, in vitro experiments suggest that oligodendrocytes are more vulnerable to slowly developing hypoxia (as seen in MS) compared to acute anoxia [40]. This is because hypoxic conditions lead to significantly more lipid peroxidation—a process driven by free radicals—indicating that oxidative damage is a major factor in their death under these conditions [40]. Moreover, hypoxia can increase intracellular calcium levels through specific ion channels, contributing further to oligodendrocyte degeneration [41].

Oligodendrocyte precursor cells (OPCs) are more vulnerable to hypoxia than adult myelinating oligodendrocytes [40,42,43]. Hypoxia inhibits OPC spinal cord and forebrain dorsal migration, delays the onset of oligodendrocyte myelination in vivo, and reduces myelin basic protein (MBP) expression in Zebrafish [44]. Oxygen tension is crucial for OPC proliferation and differentiation, and their ability to express myelin components in rat OPC and organotypic hippocampal slice cultures [45].

### 3.3. Hypoxia and Inflammation Interactions

There is an extensive crosstalk between hypoxia-inducible factors (HIF) and nuclear factor kappa-light-chain-enhancer of activated B cells (NF-κB) involved in hypoxia and inflammation, which includes common activating stimuli, shared regulators and targets [46]. Both HIF and NF-κB play significant roles in regulating inflammatory responses, immune cell activation, and tissue injury, making them key players in the disease process [46].

HIF, a master regulator for the body’s hypoxic response, is a transcription factor that regulates the expression of genes that contribute to metabolism and inflammation. HIF levels in the cells are tightly controlled by a group of hydroxylases, i.e., three prolyl hydroxylases (PHDs) and one asparagine hydroxylase—factor inhibiting HIF (FIH) [47]. HIF is stabilized and activated in hypoxic environments, triggering the transcription of genes involved in angiogenesis, metabolic adaptation, and inflammatory responses, such as erythropoietin (EPO) and vascular endothelial growth factor (VEGF) (Figure 1) [48,49]. Recombinant human EPO administration enhanced remyelination in organotypic rat spinal cord slices after demyelination induced by lysolecithin [50], and induced OPC proliferation after traumatic brain injury in mice [51]. During hypoxia, astrocytes upregulate HIF signalling, which promotes Wingless-related integration site (Wnt) ligand secretion onto endothelial cells to stimulate angiogenesis [52]. The stimulation of angiogenesis, through the upregulation of VEGF in MS, may help the repair process but also contribute to pathological neovascularization, causing BBB leakage [53]. Activation of HIF in OPCs couples postnatal white matter angiogenesis, axon integrity, and onset of myelination in the mammalian forebrain [54,55]. However, prolonged HIF activity impairs OPC function. Allen et al. [56] demonstrate that chronic accumulation of HIF-1α activates non-canonical targets to impair the generation of oligodendrocytes from pluripotent stem cell-derived OPCs.

HIF-1α is essential for the differentiation and inflammatory function of TH17 cells, which are involved in several autoimmune diseases, including MS [57,58]. In the EAE model, T cell-specific Hif-1α knockout results in the resistance to EAE, which is related to the suppression of TH17 cell development in favour of regulatory T cells (Tregs) differentiation [59]. But Clambey et al. [60] observed that under hypoxic conditions there is an increase in the abundance of Tregs, and this effect is mediated by HIF-1α, which regulates the expression of the forkhead box p3 (*FoxP3*) gene. FoxP3 is a key transcription factor necessary for the development and function of Tregs. The increased expression of FoxP3 in response to hypoxia promotes the differentiation of Tregs and strengthens their anti-inflammatory activity.

NF-κB is a transcription factor that plays a critical role in regulating immune responses and inflammation [61]. In MS, NF-κB activation is linked to the initiation and propagation of the autoimmune attack on the CNS [62]. Immunohistochemistry studies show that NF-κB activation is often co-localized with markers of inflammation (e.g., tumor necrosis factor (TNF)-α, interleukin (IL)-1β) and of tissue injury in both active and chronic lesions [63]. It enhances the production of pro-inflammatory cytokines (e.g., TNF-α, IL-1, IL-6) and adhesion molecules that drive the autoimmune response, leading to further demyelination and axonal damage [62]. NF-κB plays a pivotal role in the activation of T cells, macrophages, and microglia in MS, which, when activated, further contribute to tissue damage [64]. Dysregulated NF-κB signalling in microglia and astrocytes contributes to the chronic inflammatory state observed in MS [64]. It can perpetuate immune cell recruitment and neuroinflammation, leading to a vicious cycle of tissue damage [65]. NF-κB also affects OPCs and the remyelination process. Aberrant NF-κB signalling can impair OPC differentiation, hindering repair mechanisms and contributing to the progressive phase of MS. Histological analysis has revealed the presence of NF-κB-p65 (the active subunit of NF-κB) in MS lesions, particularly in infiltrating immune cells, such as microglia and macrophages [64].

Both HIF and NF-κB are activated in response to hypoxia and inflammation, and they have been shown to co-regulate certain genes, particularly those involved in inflammation and immune response (Figure 1). Inflammation can trigger hypoxia by damaging mitochondria and endothelial cells, impairing blood flow regulation [66]. NF-κB activation in response to pro-inflammatory cytokines can induce the expression of factors like TNF-α, IL-6, and COX-2, which can, in turn, stabilize HIF-1α even under normoxic conditions [67], further promoting the hypoxic response. Conversely, hypoxia-induced activation of HIF-1α in inflammatory cells can increase the expression of pro-inflammatory cytokines and chemokines, which activate NF-κB signalling in neighbouring cells, leading to a positive feedback loop of inflammation [68]. In hypoxic regions of MS lesions, HIF-1α and NF-κB can cooperate to promote the transcription of genes involved in immune cell recruitment, cytokine production, and tissue repair. HIF and NF-κB both regulate the expression of matrix metalloproteinases (MMPs) that degrade myelin and extracellular matrix components [69], thus facilitating demyelination in MS lesions. The co-activation of HIF-1α and NF-κB in oligodendrocytes led to increased production of pro-inflammatory mediators, contributing to oligodendrocyte apoptosis and inhibition of remyelination [70,71,72]. Hypoxia combined with IL-1β and TNF-α enhance leukocyte adhesion molecule expression in brain endothelial cell cultures, and thus a pro-inflammatory environment in combination with endothelial cell dysfunction could trigger abnormal CNS leukocyte trafficking [73].

## 4. HB-EGF Emerges as a Potential Biomarker for Progression of Multiple Sclerosis

Heparin-binding epidermal growth factor (EGF)-like growth factor (HB-EGF) is a member of the EGF family of proteins, with essential roles in tissue regeneration and neuronal survival during ischemic diseases and the development of the CNS [74]. During the early episodes of CNS inflammation in MS, astrocytes secrete HB-EGF, where it controls recovery from acute autoimmune inflammation, but HB-EGF is suppressed in the later stages of CNS inflammation [75]. HB-EGF is elevated in the cerebrospinal fluid (CSF) of CIS patients compared with RRMS patients, suggesting that CSF HB-EGF could be a potential biomarker for diagnosing CIS and discriminating between non-CIS vs. CIS patients [75]. There is a negative correlation between the number of CNS lesions and HB-EGF levels in the CSF of MS patients. HB-EGF is detected in the acute stages of MS and its presence is a vital risk factor for the conversion to RRMS. In contrast to CSF levels, HB-EGF serum levels are reduced in CIS patients compared with controls, suggesting the existence of a CNS-specific regulation in the initial stages of autoimmune CNS inflammation [75]. Linnerbauer et al. [75] showed that astrocyte-specific inactivation of *HB-EGF* via a lentiviral vector interfered with the recovery from acute EAE, while intracerebroventricular injection of IL-1β and TNF-α resulted in an increase in HB-EGF expression in an EAE mouse model.

HB-EGF exerts its effects by binding to the EGF receptor (EGFR), which is also called ERBB1 (Erythroblastic Leukemia Viral Oncogene Homolog 1). The activation of EGFR leads to receptor dimerization and activation of intracellular signalling pathways, including phosphoinositide 3-kinase (PI3K)/protein kinase B (AKT), mitogen-activated protein kinase (MAPK), and protein kinase C (PKC) (Figure 2) [76]. These signalling cascades regulate various cellular processes, including cell survival, proliferation, migration, and differentiation [76].

The HB-EGF signalling pathway plays a crucial role in the process of remyelination, through several mechanisms, particularly in the context of neuroinflammation, and the interaction between these cells and the extracellular environment [77]. HB-EGF signalling can enhance the proliferation and differentiation of OPCs into mature oligodendrocytes, the cells that are responsible for myelin formation and the myelination process [78]. HB-EGF is also involved in modulating inflammation [79]. By regulating the inflammatory microenvironment, it may promote a more favourable condition for oligodendrocyte differentiation and remyelination. HB-EGF also promotes tissue repair and remodelling in the CNS by influencing the extracellular matrix components and promoting cell migration [74]. This is important for the dynamic process of remyelination, as OPCs must migrate to areas of demyelination and participate in the formation of new myelin sheaths.

In addition to its influential role on oligodendrocytes, EGFR signalling has important implications for axonal growth and neuronal function. Signals from astrocytes, modulated by EGFR activation, can influence axon elongation and dendrite branching [80]. Studies have shown that phosphorylation induced by EGF stimulates neurite outgrowth in neuronal models, such as PC12 cells and cortical neurons [81,82].

HB-EGF secretion happens through a process that involves its synthesis, cleavage by ADAM (a disintegrin and metalloproteinase) proteases, and release into the extracellular space where it can mediate cell signalling by interacting with receptors on target cells [83]. The release of HB-EGF can be regulated by various extracellular signals and conditions, such as inflammatory cytokines, growth factors, and other signalling molecules, often in response to changes in the cellular environment (Figure 2) [76]. For instance, HB-EGF is found in brain cells and its expression is increased after hypoxic or ischemic injury, primarily mediated through the activation of HIF, which also stimulates neurogenesis [84]. Under low oxygen conditions, HIF-1α can activate the transcription of genes that lead to the upregulation of HB-EGF [85]. The HB-EGF gene contains promoter regions that may respond to hypoxic stimuli via HIF-mediated transcriptional activation [86]. When HIF-1α is stabilized under hypoxia, it could upregulate the transcription of HB-EGF, promoting its secretion [85,86]. Hypoxic stress may upregulate the expression or activation of ADAM17 or other proteases in astrocytes, leading to an increase in the proteolytic release of soluble HB-EGF from the cell surface into the extracellular environment [87]. Hypoxia often occurs in concert with inflammation, especially in conditions such as MS [88]. Cytokines and other inflammatory signals released during hypoxia can further stimulate HB-EGF secretion. For example, the pro-inflammatory cytokine TNF-α can enhance the activity of ADAM17, contributing to the shedding of HB-EGF from the cell surface [89].

## 5. Current and Emerging Therapies for Multiple Sclerosis

Although there is no cure for MS, there are many new treatment options aimed at slowing its progression, decreasing its symptoms, and improving mental and physical functions. There are now available drugs that are designed to reduce the risk of relapses and the formation of new MS plaques in the CNS, as well as slow the progression of disability and the loss of brain volume mass. Corticosteroids like intravenous methylprednisolone (Depo-Medrol^®^ and Medrol^®^) are used for managing MS relapses. These medications help reduce inflammation, a key factor in MS attacks, and can improve recovery from relapses. Another treatment option is plasmapheresis, a procedure that involves removing the plasma portion of the blood, cleansing it of circulating harmful proteins, and then returning it to the body. This can also aid in the recovery process, particularly during severe or refractory relapses. The modified Story Memory Technique has been developed to address the cognitive challenges in MS [90]. This approach helps individuals retain new memories and recall information by associating the information with a story that uses imagery and context. By improving memory retention and recall, this method offers an important tool for people with MS who face cognitive difficulties [90].

### 5.1. Food and Drug Administration Approved Drugs

Most drugs approved by the FDA since the early 1990s are effective at helping to manage RRMS but have little impact on SPMS (Table 1). Therefore, the best strategy for the clinician is to develop a treatment regimen during the earlier relapsing–remitting phase of the disease.

Dimethyl fumarate (Tecfidera^®^) and diroximel fumarate (Vumerity^®^) are both prodrugs of monomethyl fumarate (Bafiertam^®^). In preclinical studies, they activated nuclear factor erythroid 2-related factor 2 (Nrf2), a transcriptional factor that is up-regulated under oxidative stress [91]. Siponimod (Mayzent^®^), a sphingosine 1-phosphate receptor modulator, prevents certain white blood cells in the lymph nodes from entering the CNS [92]. Cladribine (Mavelclad^®^) is a purine analogue that suppresses lymphocytes, the cells that drive the immune response associated with MS [93]. B-cell therapies target B cells to help decrease relapses and slow the progression of MS [94]. The FDA has approved two such medications for the treatment of relapsing forms of MS, including ocrelizumab (Ocrevus^®^) and ofatumumab (Kesimpta^®^). Ocrevus^®^ (ocrelizumab) is the only FDA-approved treatment for PPMS, helping to reduce relapses and slow disability progression [95]. It is also approved for treating other forms of MS, including CIS, RRMS, and SPMS (Table 1). Dalfampridine (Ampyra^®^), in contrast, is not a disease-modifying therapy (DMT) but rather a medication aimed at improving walking ability in people with MS by blocking potassium channels. While it does not slow disease progression, it can be used alongside other DMTs to help with mobility issues [96]. Natalizumab (Tysabri^®^) is a monoclonal antibody that decreases relapse rates, slows disability progression in people with MS, and is particularly effective in reducing disease activity in RRMS [97]. Alemtuzumab (Lemtrada^®^) is another treatment for RRMS. While its exact mechanism of action is not fully understood, it is believed to target the CD52 protein expressed on immune cells, leading to their destruction [98]. Initially approved for leukaemia at higher doses, it is now used at a lower dose to treat MS.

DMT treatment for MS typically continues indefinitely unless a person experiences a poor response, intolerable side effects, or fails to adhere to the prescribed regimen. Treatment may also be adjusted if a more effective option becomes available. Decisions about ongoing treatment depend on individual responses and evolving treatment options.

### 5.2. Recent Developments and Emerging Therapies

Multiple approaches are being explored in the field of MS treatment, focusing on improving the effectiveness of immunomodulatory therapies and exploring ways to protect and regenerate the nervous system. There are already approved DMTs with immunomodulatory drugs, as listed in Table 1. There is continuous development of novel immunomodulatory agents with better safety profiles, fewer side effects, and more effective results (Table 2). Remyelination is one of the most promising therapeutic avenues for MS and researchers are developing agents to promote the regeneration of this protective myelin layer, e.g., stem cell therapy or drugs that stimulate the production of OPCs and enhance the differentiation of these cells into mature, myelinating cells (Table 2). Certain molecules like neuregulin-1, brain-derived neurotrophic factor (BDNF), and HB-EGF are being explored for their potential to promote remyelination (Table 2). Neuroprotective therapies are focused on reducing neurodegeneration due to demyelination and/or direct damage to neurons, slowing the progression of disability, and preserving cognitive function (Table 2).

## 6. Conclusions and Future Perspectives

Inflammation is the initial event in MS, causing activation of immune cells such as T cells and macrophages, destroying myelin sheath and damaging nerves. The activation of NF-κB is one of the molecular mechanisms that sustain and amplify this inflammation. In addition, hypoxia is found in several regions in the CNS of MS patients, and demyelination lesions tend to form in regions that are sensitive to hypoxia [130]. Beyond the clear potential for hypoxic conditions to exacerbate MS progression, it is highly possible that persistent hypoxia within lesions may impair the proliferation, migration, and/or differentiation of OPCs, impairing remyelination. There is an extensive crosslink between HIF and NF-κB involved, respectively, in hypoxia and inflammation. HIF stabilisation and activation is the body’s response to hypoxia. It is neuroprotective and could potentially be used to manage metabolic disturbances in MS patients, particularly those with fatigue, a common symptom of MS. On the other hand, hypoxia-induced activation of HIF-1α in inflammatory cells leads to NF-kB activation in neighbouring cells through upregulation of the expression of proinflammation cytokines and chemokines. While NF-κB mediates immune cell activation and chronic inflammation, in turn, it drives brain regional hypoxia. The interaction between these two pathways amplifies the pathological processes that drive MS, making them potential therapeutic targets for managing the disease. HB-EGF has recently emerged as a potential biomarker for identifying MS progression. Directly applying HB-EGF in EAE mice improved recovery following demyelination in preclinic studies [75]. The HB-EGF signalling pathway might be a novel target for preventing MS progression. Most of the current FDA-approved drugs are immunomodulators, which are still the cornerstone for treating MS. Nevertheless, immunomodulators do not cure the disease; ultimately, axonal injuries and neural loss lead to symptom development. Therefore, future therapies will need to focus on remyelination and neuroprotection in addition to immunomodulation. Remyelination has the potential to reverse MS symptoms and slow or even prevent disease progression, while neuroprotection aims at protecting neurons from damage and loss, restoring nerve conduction and metabolic support to axons. Combined therapies that target multiple aspects of the disease, based on the patient’s disease characteristics, could potentially revolutionize the way MS is treated in the coming years. Understanding how hypoxia contributes to MS progression, including through limited remyelination, warrants greater prioritisation of research on this aspect of the disease. Promoting the brain’s endogenous neuroprotection by targeting HIF and HB-EGF could provide new therapies for MS patients.

## Figures and Tables

**Figure 1 brainsci-15-00248-f001:**
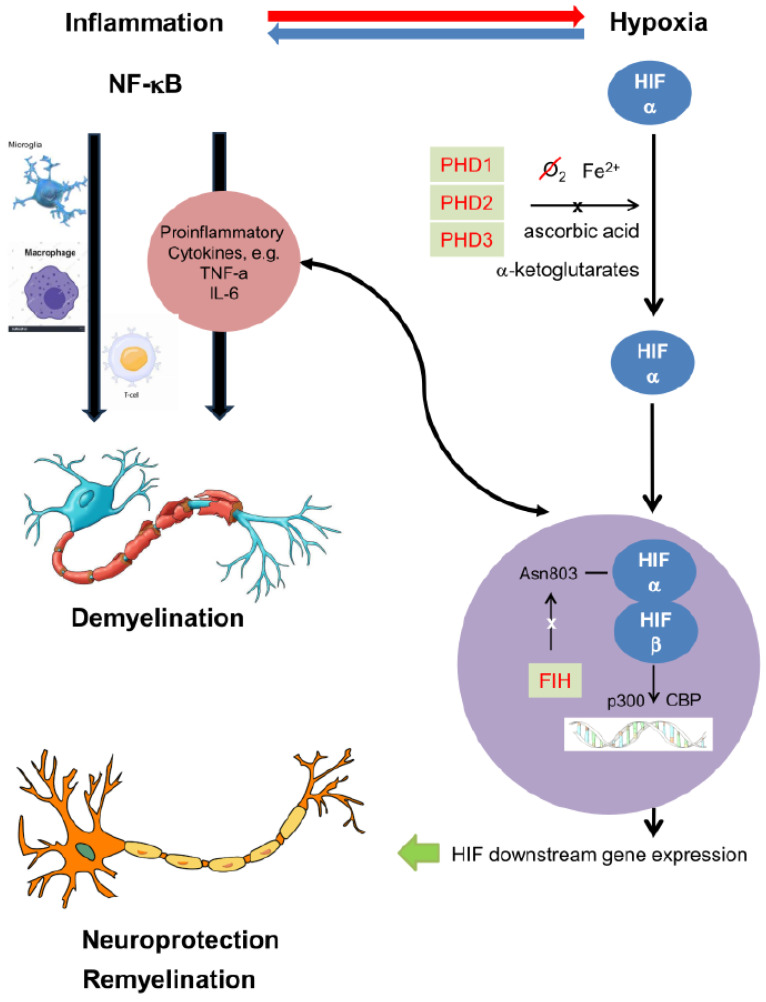
The crosslink between HIF and NF-kB in hypoxia and inflammation, respectively, seen in multiple sclerosis. In multiple sclerosis (MS), immune cell (like T-cells and macrophages) activation attacks the myelin sheath of axons, leading to demyelination and neuronal damage. The activation of NF-κB occurs as part of the inflammatory process, driving the production of pro-inflammatory cytokines and chemokines, which further exacerbate the disease. Inflammation can additionally trigger hypoxia by damaging mitochondria and endothelial cells, impairing blood flow regulation and leading to the stabilization of HIF. HIF activation upregulates hundreds of human genes contributing to neuroprotection and remyelination, but also producing proinflammatory cytokines, further activating NF-κB. The combined actions of HIF and NF-κB can influence cellular processes such as survival, apoptosis, angiogenesis, and metabolism, and have significant roles in demyelination, remyelination, and neuroprotection in MS.

**Figure 2 brainsci-15-00248-f002:**
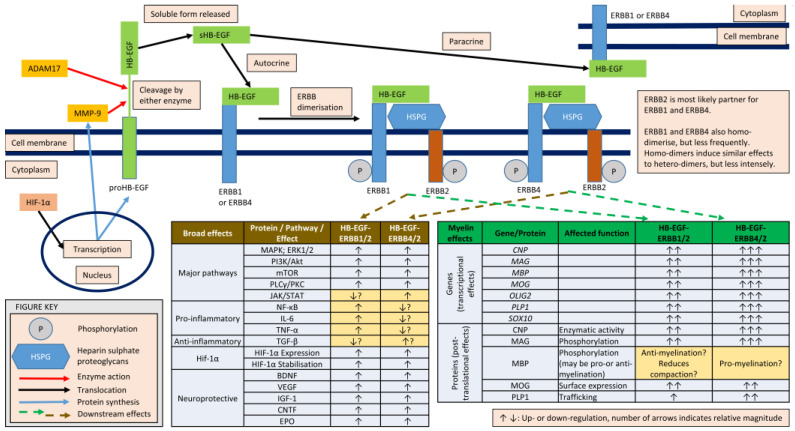
The molecular cascades that follow the release of HB-EGF and activation of ERBB receptor dimers, with relevance to neuroprotection and remyelination. Pro-HB-EGF is membrane-bound in all major CNS cell types. Multiple enzymes (commonly ADAM17 and MMP-9) cleave pro-HB-EGF, releasing soluble HB-EGF (sHB-EGF) extracellularly. This release is increased within MS lesions and under hypoxic conditions. HB-EGF may act in an autocrine or paracrine fashion, binding ERBB1 or ERBB4 receptors, which dimerise, typically with ERBB2. These dimers then induce a wide variety of intracellular effects, as indicated in the tables. The arrows inside tables indicate increasing (up) or deceasing (down) intracellular effects. All neural cell types express these receptors, but the table related to myelin effects (green headings) shows molecular changes that are specific to cells of the oligodendroglial lineage. Many effects are similar for both dimer forms but there may be some differences, indicated by the light beige colour of the Table cells. Most of these effects would be considered cytoprotective and/or pro-repair, particularly in the context of MS lesions, where remyelination would be desirable, or under hypoxic stress. Note that HIF-1α activity, upregulated in hypoxia, increases the release of sHB-EGF, while activation of ERBB dimers stabilises HIF-1α protein and upregulates HIF-1α transcription, providing a positive feedback loop. ADAM17—a disintegrin and metalloproteinase 17; ERBB—Erythroblastic Leukemia Viral Oncogene Homolog (alternative names, ERBB1 = EGFR; ERBB2 = HER2; ERBB4 = HER4); HB-EGF—heparin-binding EGF-like growth factor (EGF—epidermal growth factor); HIF-1α—hypoxia-inducible factor 1α; HSPG—heparin sulphate proteoglycans; MMP-9—matrix metalloproteinase 9. The arrows inside tables indicate increasing (up) or deceasing (down) intracellular effects. Question marks within the Tables would be valuable in indicating the more speculative signalling effects.

**Table 1 brainsci-15-00248-t001:** The list of FDA-approved drugs used for MS patients.

Brand Name (Generic Name)	Approval	Mechanisms of Action	Indications
	Date		
Ampyra (dalfampridine)	22 January 2010	Potassium channels blocker	Symptom management therapy to improve walking ability
Aubagio	12 September 2012	Pyrimidine synthesis inhibitor	CIS, RRMS, SPMS
(teriflunomide)			
Avonex	17 May 1996	Immunomodulator	CIS, RRMS, SPMS
(interferon beta-1a)			
Bafiertam	28 April 2020	Nrf2 pathway activator	CIS, RRMS, SPMS
(monomethyl fumarate)			
Briumvi	28 December 2022	CD20-directed cytolytic monoclonal antibody	CIS, RRMS, SPMS
(ublituximab-xiiy)			
Copaxone	20 December 1996	Immunomodulator	CIS, RRMS, SPMS
(glatiramer acetate)			
Extavia	14 August 2009	Immunomodulator	CIS, RRMS, SPMS
(interferon beta-1b)			
Gilenya	21 September 2010	Sphingosine 1-phosphate receptor modulator	CIS, RRMS, SPMS
(fingolimod)			
Kesimpta	20 August 2020	CD20-directed cytolytic monoclonal antibody	CIS, RRMS, SPMS
(ofatumumab)			
Lemtrada	14 November 2014	CD52-directed cytolytic monoclonal antibody	CIS, RRMS, SPMS
(alemtuzumab)			
Mavenclad	29 March 2019	Purine antimetabolite	CIS, RRMS, SPMS
(cladribine)			
Mayzent	26 March 2019	Sphingosine-1-phosphate receptor modulator	CIS, RRMS, SPMS
(siponimod)			
Ocrevus	28 March 2017	CD20-directed cytolytic monoclonal antibody	CIS, RRMS, SPMS, PPMS
(ocrelizumab)			
Plegridy	15 August 2014	Immunomodulator	CIS, RRMS, SPMS
(peginterferon beta-1a)			
Ponvory	18 March 2021	Sphingosine 1-phosphate receptor modulator	CIS, RRMS, SPMS
(ponesimod)			
Rebif	3 January 2013	Immunomodulator	CIS, RRMS, SPMS
(interferon beta-1a)			
Tecfidera	27 March 2013	Nrf2 pathway activator	CIS, RRMS, SPMS
(dimethyl fumarate)			
Tyruko	25 August 2023	A ‘biosimilar medicine’ of Tysabri	CIS, RRMS, SPMS
(natalizumab)			
Tysabri	23 November 2004	Integrin receptor antagonist	CIS, RRMS, SPMS
(natalizumab)			
Vumerity	29 October 2019	Nrf2 pathway activator	CIS, RRMS, SPMS
(diroximel fumarate)			
Zeposia	25 March 2020	Sphingosine 1-phosphate receptor modulator	CIS, RRMS, SPMS
(ozanimod)			
Zinbryta (withdrawn from the market in March 2018)	27 May 2016	IL-2 receptor blocking antibody	CIS, RRMS, SPMS

**Table 2 brainsci-15-00248-t002:** Potential agents developed for treating MS.

Agents	Mechanism of Action
	i. immunomodulation
Bruton’s tyrosine kinase (BTK) inhibitor	By targeting BTK, these inhibitors aim to reduce the activity of B cells and microglia, potentially reducing inflammation and preventing further damage to the nervous system. Although BTK inhibitors are still being studied, they represent a promising avenue of therapy, particularly for those with RRMS and SPMS [99].
Laquinimod	Laquinimod has multiple mechanisms of action that involve modulating immune cells and glial cells to reduce inflammation, promote anti-inflammatory pathways, and protect against neurodegeneration, making it a promising candidate for MS treatment [100].
Tetra methyl pyrazine	Tetra methyl pyrazine shifts microglia toward an anti-inflammatory state, along with its effects on signal transducer and activator of transcription 3 (STAT3)/suppressor of cytokine signalling 3 (SOCS3) and NF-kB signalling, helps preserve BBB integrity and prevent the damage associated with demyelination in EAE mice [101].
Statins	Statins may have a role in reversing and preventing relapsing and chronic forms of EAE mice [102,103], highlighting the potential immunomodulatory effects of statins beyond their well-known cholesterol-lowering properties.
	**ii. remyelination**
Myelin peptides	Transdermal myelin peptides significantly reduced MS lesions and relapses, with no serious adverse events [104].
Anti-LINGO-1 antibodies	In Phase 2 clinical trials, anti-LINGO-1 antibodies were tested in individuals who experienced their first episode of optic neuritis, an inflammatory condition often associated with MS that affects the optic nerve. The results showed improvement in nerve impulse conduction along the affected optic nerve, suggesting that the treatment may enhance myelination and improve nerve function [105]. This study demonstrates that targeting LINGO-1 can not only promote OPC differentiation and myelination but also potentially lead to functional improvements in MS.
Benztropine	Antagonism of M1/M3 muscarinic acetylcholine receptors to stimulate oligodendrocyte differentiation. Deshmukh et al. [106] found Benztropine enhanced remyelination in experimental MS models.
Clobetasol and Miconazole	Najm et al. [107] demonstrated that the treatment with Clobetasol and Miconazole enhanced remyelination and promoted oligodendrocyte differentiation.
Guanabenz	Guanabenz is an α2 adrenergic receptor agonist. It prevents myelin loss and increases oligodendrocyte survival as well as reduces deficits in the EAE model [108].
Clemastine fumarate	Clemastine fumarate is an over-the-counter antihistamine. It has demonstrated an ability to repair myelin in patients with MS [109].
neuregulin-1 beta 1	Neuregulin-1 beta 1 (Nrg-1β1) is involved in the signalling pathways that regulate the development and maintenance of the nervous system, including the myelination of axons by oligodendrocytes. Its disruption may affect myelin integrity and neuroprotection, potentially exacerbating MS pathology [110]. Studies in EAE mice have shown promising results with pre-symptomatic administration of Nrg-1. When administered before the onset of symptoms, Nrg-1 significantly delayed the appearance of disease symptoms. Moreover, when symptoms did eventually manifest, they were less severe, indicating that Nrg-1 could offer neuroprotective effects and possibly slow disease progression [111].
BNDF	BDNF has been found to be neuroprotective as well as a key molecule controlling remyelination in MS [112,113].
HB-EGF	Linnerbauer et al. [75] conducted a study in which recombinant mouse HB-EGF (rmHB-EGF) was administered intranasally to EAE mice after the onset of symptoms. The treatment led to significant improvements in recovery and reduced peripheral immune cell infiltration into the spinal cord.
Stem cells	Leone et al. [114] injected foetal allogeneic neural stem cells in the brain ventricles of SPMS patients and found that the stem cells were well tolerated and had a long-lasting effect that appeared to protect the brain from further damage.
	**iii. neuroprotection**
Anti-ASIC-1 (acid-sensing ion channel 1)	Inhibiting ASIC-1 helps to prevent the intracellular accumulation of Ca^2+^ and Na^+^, which are toxic in excess and can lead to cell death and tissue damage. This inhibition can protect neurons and glial cells in MS lesions, contributing to reduced brain atrophy—a common feature of MS progression [115].
Nimodipine	Treatment with nimodipine (a voltage-gated Ca^2+^ channel blocker) in EAE rats improved spinal function, restored oxygenation, and reduced demyelination [116].
Lamotrigine	Lamotrigine increases axonal conduction and protects dorsal column white matter in EAE mice [117]. In a clinical trial in SPMS, lamotrigine treatment significantly reduced serum neurofilament level and disability [118].
Phenytoin	Phenytoin protects spinal cord axons from degeneration and improves outcomes in EAE mice [119].
Riluzole	Riluzole reduces the amount of non-phosphorylated neurofilament in EAE mice, thus providing evidence of reduced demyelination [120].
Tacrolimus (FK506) and FK1706	The non-immunosuppressive derivatives FK1706 and FK506 (immunosuppressive drugs used for organ transplantation) reduced spinal cord white matter injury in the rodent EAE model [121,122].
2,3-dihydroxy-6-nitro-7-sulfamoyl-benzo (f)-quinoxaline-2,3-dione (NBQX)	A kainite/AMPA antagonist that reduces axonal injury and increases oligodendrocyte survival in EAE mice [123].
	**iv. oxygen therapies or HIF modulation**
Ozone autohemotherapy (O_3_-AHT)	O_3_ gas prevents free radical damage, induces oxidative stress tolerance and has antioxidant capabilities in MS patients. O_3_ therapy increased the expression levels of FoxP3, miR-146a, TGF-β and IL-10 cytokines, and elevated the frequency of Treg cells [124,125].
Remote ischemic-preconditioning (RIPC)	RIPC in hindlimb increased mRNA and serum expression of HIF1-α and EPO in the brain of EAE mice [126]. RIPC was shown to improve walking distance as measured by the six-minute walk test in a clinical trial in patients with MS. The RIPC group demonstrated a 5.7% improvement in distance walked compared to the placebo group [127].
VCE-004.8, a cannabidiol quinone derivative	VCE-004.8 stabilises HIF-1α and HIF-2α in human oligodendrocyte and microglial cells, modulates neuroinflammatory activity, enhances remyelination, and induces expression of neuroprotective factors such as EPO and VEGF [128].
Acriflavine and echinomycin	Acriflavine is a topical antiseptic that can inhibit HIF. It was found to “preserve vision” in EAE, although echinomycin (a peptide antibiotic intercalating into DNA, thereby blocking the binding of HIF-1α) did not have the effect [129].
NtHIF-1α-TMD	NtHIF-1α-TMD blocks the differentiation of naive T cells into TH17 cells in a dose-dependent manner in an in vitro setting, but also causes TH17 cells to adopt an immune-suppressive phenotype in EAE mice [57], possibly reducing inflammation or autoimmunity.

## Data Availability

Not applicable.

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
