# Peer review of "Hypoxic Neuroinflammation in the Pathogenesis of Multiple Sclerosis"

_brainsci, 2025, doi:10.3390/brainsci15030248_

Round 1

Reviewer 1 Report

Comments and Suggestions for Authors

Dear Authors,

The review reveals the cellular and molecular basis of demyelination in multiple sclerosis. The role of typical pathological processes is described: inflammation, hypoxia and their interaction. The participation of different cells of the nervous tissue in the development of pathological disorders in multiple sclerosis is shown. Unfortunately, the review is poorly structured, the information is presented extensively. Obviously, the authors want to show the mechanism of pathogenesis associated with the participation of neuroinflammation and hypoxia. The review mainly presents known information, including information on neuroinflammation and hypoxia in multiple sclerosis. In this regard, it is necessary to explain the novelty, originality and necessity of this review.

The aim of the review article is to investigate the neuroinflammation and hypoxia cycle in the pathogenesis of MS and identify therapeutic targets to break the neuroinflammation and hypoxia cycle.

The objective of the review, which is stated in the abstract, should also be presented and justified in the maintext of the review.

It is necessary to indicate the type of review and, accordingly, the methodology for searching and rating publications.

The drawings are decorative, but they represent the pathological process in multiple sclerosis too simplified. The cause-and-effect relationships shown in the drawings should be carefully checked.

The "Treatment" section presents groups of drugs recommended for the treatment of MS. However, the section requires revision, it is necessary to present the effectiveness and evidence of the effects of the proposed approaches to metabolic therapy of MS.

The abstract needs improvement. It is necessary to more clearly present the concept of brain damage in MS and the rationale for pharmacological correction of metabolic disturbances.

Author Response

Thank you very much for your critical comments on our manuscript. We have made changes accordingly and provided point-to-point responses as follows:

Unfortunately, the review is poorly structured, the information is presented extensively.

Reply: We have rewritten the manuscript and re-structured the paper, and removed all unnecessary parts to make the paper concise and focused.

The review mainly presents known information, including information on neuroinflammation and hypoxia in multiple sclerosis. In this regard, it is necessary to explain the novelty, originality and necessity of this review.

Reply: We have added a paragraph of “HB-EGF emerges as a potential biomarker for progression of multiple sclerosis” to present new discoveries in the studies of multiple sclerosis. We have also made fresh comments on inflammation and hypoxia interactions and updated the section of “current and emerging therapies for multiple sclerosis”.

The objective of the review, which is stated in the abstract, should also be presented and justified in the maintext of the review.

It is necessary to indicate the type of review and, accordingly, the methodology for searching and rating publications.

Reply: We have stated and justified the aim of the review article with the indication of the review type in the main text as follows:

“The aims of this narrative review are to investigate the roles of neuroinflammation and hypoxia interaction in the MS progression, and to identify emerging biomarkers for MS progression.”

The drawings are decorative, but they represent the pathological process in multiple sclerosis too simplified. The cause-and-effect relationships shown in the drawings should be carefully checked.

Reply: We have redrawn the figures and reduced the figures from 5 to 2.

The "Treatment" section presents groups of drugs recommended for the treatment of MS. However, the section requires revision, it is necessary to present the effectiveness and evidence of the effects of the proposed approaches to metabolic therapy of MS.

Reply: We revised this section and added a paragraph for future perspective on the treatment to present evidence of the effects of the proposed approaches to prevent multiple sclerosis from progression. Thanks for the critical comments.

The abstract needs improvement. It is necessary to more clearly present the concept of brain damage in MS and the rationale for pharmacological correction of metabolic disturbances

Reply: We revised the abstract and clearly presented the rational for proposed new treatments for multiple sclerosis.  

Reviewer 2 Report

Comments and Suggestions for Authors

I would like to start by congratulating the authors on their review titled “Hypoxic Neuroinflammation in the Pathogenesis of Multiple Sclerosis”.

I only have a few remarks:
The section on the symptoms and types of MS has some weaknesses:

  • Detailed but redundant descriptions. Much of the information is already well-known and does not seem directly relevant to the objectives of the study.
  • There is a lack of critical comparisons between the types of MS and their implications in the hypoxia-neuroinflammation pathogenesis.

The section on Interactions between hypoxia and inflammation, given that the focus is the role of hypoxia in MS, discusses the bidirectional relationship in a rather generic way and lacks details about key molecular pathways. Additionally, illustrations such as Figure 4 are not accompanied by in-depth analysis.

I would therefore suggest describing the role of HIF and NF-κB in detail, with concrete examples of studies demonstrating their importance in MS, and expanding the discussion on histological evidence of hypoxia in MS patients.

Finally, I recommend strengthening the rationale by providing an overview of pre-existing evidence on the link between hypoxia and MS. Additionally, clearly highlighting the innovative purpose of the study would be beneficial.

Author Response

Thank you very much for your critical comments on our manuscript. We have made changes accordingly and provided point-to-point responses as follows:

Detailed but redundant descriptions. Much of the information is already well-known and does not seem directly relevant to the objectives of the study.

Reply: Thanks for the comments. We have removed all unnecessary and redundant descriptions parts, making paper concise and focused.

There is a lack of critical comparisons between the types of MS and their implications in the hypoxia-neuroinflammation pathogenesis.

Reply: We have made a new paragraph of “progressive nature of multiple sclerosis” to compare between types of multiple sclerosis.

The section on Interactions between hypoxia and inflammation, given that the focus is the role of hypoxia in MS, discusses the bidirectional relationship in a rather generic way and lacks details about key molecular pathways. Additionally, illustrations such as Figure 4 are not accompanied by in-depth analysis.

Reply: We have rewritten the paragraph of “neuroinflammation and hypoxia cycle in multiple sclerosis” to discuss the bidirectional relationship with details on key signalling pathways. We have made a new figure on the crosslink between HIF and NF-kB with depth analysis.

I would therefore suggest describing the role of HIF and NF-κB in detail, with concrete examples of studies demonstrating their importance in MS, and expanding the discussion on histological evidence of hypoxia in MS patients.

Reply: We have discussed the roles of HIF abd NF-kB in detail in paragraph 3.3. “hypoxia and inflammation interaction”.

Finally, I recommend strengthening the rationale by providing an overview of pre-existing evidence on the link between hypoxia and MS. Additionally, clearly highlighting the innovative purpose of the study would be beneficial.

Reply: We have strengthened the rationale by providing an overview of pre-existing evidence on the link between hypoxia and MS and highlighted the innovative purpose of the study. Many thanks for your critical comments.

Round 2

Reviewer 1 Report

Comments and Suggestions for Authors

the manuscript has been sufficiently improved to warrant publication in Brain Sciences.

Formatting according to template is required.